# Formation of Supramolecular Structures in Oxidation Processes Catalyzed by Heteroligand Complexes of Iron and Nickel: Models of Enzymes

**DOI:** 10.3390/ijms26168024

**Published:** 2025-08-19

**Authors:** Ludmila Ivanovna Matienko, Elena M. Mil, Anastasia A. Albantova, Alexander N. Goloshchapov

**Affiliations:** Institute of Biochemical Physics, Russian Academy of Sciences, 4 Kosygin str., Moscow 119334, Russia

**Keywords:** supramolecular structures, intermolecular regulatory interactions, H-bonds, self-assembly, homogeneous and enzymatic catalysis, model heteroligand Ni(Fe) complexes, AFM

## Abstract

In some cases, the catalytic processes involve the formation of self-organized supramolecular structures due to H-bonds and other non-covalent interactions. It has been suggested that the construction of self-assembled catalytic systems is a promising strategy to mimic enzyme catalysis at the model level. As a rule, the real catalysts are not the primary catalytic complexes, but rather, those that are formed during the catalytic process. In our earlier works, we have established that the effective catalysts M(II)*_x_*L^1^*_y_*(L^1^_ox_)_z_(L^2^)*_n_*(H_2_O)*_m_* (M = Ni, Fe, L^1^ = acac^−^, L^2^ = activating electron-donating ligand) for the selective oxidation of ethylbenzene to α-phenyl ethyl hydroperoxide are the result of the transformation of primary (Ni(Fe)L^1^)_x_(L^2^)_y_ complexes during the oxidation of ethylbenzene. In addition, the mechanism of the transformation to active complexes is similar to the mechanism of action of NiFeARD (NiFe-acireductone dioxygenase). Based on kinetic and spectrophotometric data, we hypothesized that the high stability of effective catalytically active complexes may be associated with the formation of stable supramolecular structures due to intermolecular hydrogen bonds and possibly other non-covalent bonds. We confirmed this assumption using AFM. In this work, using AFM, we studied the possibility of forming supramolecular structures based on iron complexes with L^2^-crown ethers and quaternary ammonium salts, which are catalysts for the oxidation of ethylbenzene and are models of FeARD (Fe-acireductone dioxygenase). The formation of supramolecular structures based on complexes of natural Hemin with PhOH and L-histidine or Hemin with L-tyrosine and L-histidine, which are models of heme-dependent tyrosine hydroxylase and cytochrome P450-dependent monooxygenases (AFM method), may indicate the importance of outer-sphere regulatory interactions with the participation of Tyrosine and Histidine in the mechanism of action of these enzymes.

## 1. Introduction

The creation of self-assembling catalytic systems was envisioned as a strategy for modeling enzymatic systems. It was assumed that the organization of several catalytic centers by noncovalent interactions into a specific framework could increase the reactivity, selectivity, and efficiency similar to the action of enzymes [1,2,3,4,5,6,7].

In recent years, various approaches have been proposed to achieve the goals of catalysis. Supramolecular strategies involving noncovalent interactions have been found to create the bases of supramolecular catalysts from initial building blocks.

For example, the authors of [8] used a cavity-containing tetrahedral metal ligand (M_4_L_6_) as a catalyst for the 3-aza-cope rearrangement of allyl-enammonium cations (Figure 1). The self-assembling cage accepts cationic guests rather than neutral guests because the cage is negatively charged.

It was established that after binding, the rearrangement rate increases up to 850 times. The supramolecular structure leads to a decrease in the entropy and enthalpy barriers to rearrangement and is very sensitive to small structural changes in the substrate [8].

A supramolecular strategy was proposed in which the encapsulation of the hydrogenation catalyst makes the regio-selective hydrogenation of olefins possible in the case of multiple sites of unsaturation [9].

In recent years, various supramolecular strategies have been developed in which noncovalent interactions lead to rapid bond formation. Breit et al. [10] reported the creation of an in situ library of bidentate ligands from the tautomeric pair 2-pyridone/2-hydroxypyridine. Dimerization of the tautomeric pair is regulated by hydrogen bonding and occurs in aprotic solvents. The rhodium catalyst formed by the self-assembly of pyridine and hydroxypyridine tautomers of 6-(diphenyl-phosphino)pyridin-2(1H)-one (6-DPPon) catalyzes the hydroformylation of a variety of structurally diverse terminal olefins, including arenes, alcohols, aldehydes, acetals, amides, esters, ethers, and carbamates.

In Ref. [11], the authors reported a dynamic catalyst that is a supramolecular complex involving H bonds between Mn(III) salen and Zn(II) porphyrin. The salen sub-unit acts as the catalytic center for the catalytic epoxidation of olefins, while the Zn-porphyrin component is the binding site. Based on the principle of nucleotide H-binding, dimeric and trimeric porphyrins were created that bound self-complementarily [12].

The catalytic activity of metalloporphyrins self-organized into nanoaggregates with diameters of 10–100 nm was found to increase in the epoxidation reaction of cyclohexene in solution compared to individual porphyrin (~70-fold increase in turnover number (TN) and 10-fold increase in rate) [13].

H-bonding was used for the construction of catalytic multinuclear systems in solution via self-assembly of monomeric forms. For example, self-organization of the dinuclear Co(II)-salen complex due to complementary hydrogen bonds leads to a growth in the catalyst activity compared with mononuclear complex: a significant increase in the reaction rate and enantioselectivity of the Henry reaction were observed [14].

The mechanism of catalysis often involves the formation of a supramolecular assembly during the reaction. In particular, this is because self-organizing structures are thermodynamically more favorable compared to oligomeric or polymeric structures [2].

Self-assembly is considered as a subset of self-organization. Specific intermolecular interactions are directional motifs designed to recognize complementary components to create intermolecular interactions such as H-bonds, non-covalent, and dipole–dipole interactions. These lead to predictable self-assembly architectures in solution. Non-specific intermolecular interactions are usually non-directional, such as dispersion forces and ionic interactions. They usually lead to structures that are difficult to predict [15,16,17].

Jean-Marie Lehn (Nobel laureate) first coined the term “supramolecular chemistry” in 1978. According to Jean-Marie Lehn's definition, “supramolecular chemistry” is the “chemistry describing complex formations that result from the association of two (or more) chemical particles bound together by intermolecular forces. Supramolecular chemistry provides ways and means for progressively unraveling the complexification of matter through self-organization” [16].

Alfred Werner was the first to express the idea of the need to take into account the influence of the microenvironment of metal complexes of variable valence on the structure and properties of the complexes [18]. The influence of H-bonds in the second coordination sphere of metal complexes on the ability of the complexes to bind and activate O_2_ was studied in [18,19]. In the case of the activation of O_2_, upon its binding to the active center of metalloenzymes, secondary interactions (hydrogen bonding, proton transfer) play an important role [20].

The creation of highly efficient, environmentally friendly industrial processes for the oxidation of hydrocarbons is largely determined by the ability of researchers to control these processes.

We first proposed the method of modification of homogeneous catalysts (Ni- or Fe-complexes) by using additions of mono- or multidentate ligands aimed at increasing the activity and effectivity of catalysts [1]. The mechanism of action of modifier ligands has been established, and new effective and selective Ni-catalysts for the oxidation of ethylbenzene into α-phenyl ethyl hydroperoxide (PEH) and cumene into cumene hydroperoxide have been developed. These hydroperoxides are intermediate products of the large-scale production of propylene oxide and styrene by the oxidation of ethylbenzene, and phenol and acetone by the oxidation of cumene.

The mechanism of action of L^2^ ligand-modifiers is associated with the formation of active primary complexes of (Ni(Fe)L^1^)_x_(L^2^)_y_ (L^1^=acac^−^) (macrostage I of oxidation). During the second macrostage of oxidation, Ni(Fe)*_x_*L^1^*_y_*(L^1^_ox_)*_z_*(L^2^)*_n_*•H_2_O_m_ heteroligand complexes are formed, which are intermediate products of the O_2_ oxidation of primary (M(II)L^1^_2_)*_x_*(L^2^)*_y_*complexes. In this case, the mechanism of catalysis changes. The by-products AP (acetophenone) and MPC (methyl phenyl carbinol) are formed not during the decay of PEH (stage I), but in parallel with PEH, namely, in the stages of chain propagation {Cat + RO_2_^•^→} and the quadratic termination of the chain {RO_2_^•^ + RO_2_^•^→}. The mechanism of oxidative transformation of complexes of (Ni(Fe)L^1^)_x_(L^2^)_y_ is similar to the mechanism of action of Ni(Fe)-acireductone dioxygenase (Ni(Fe)ARD) [21] and Fe-acetylacetone dioxygenase Dke1 [22]. Ni(or Fe)(OAc)_2_ formed at macrostage III of ethylbenzene oxidation (the final product of the transformation (Ni(Fe)L^1^)_x_(L^2^)_y_) catalyzes the heterolytic decomposition of PEP into PhOH and acetaldehyde, and the selectivity of S_PEH_ decreases. The phenomenon of the synergetic increase in parameters, namely, the conversion *C*, selectivity ***S***_max_, and [PEH]_max_, when phenol was introduced into the catalytic binary system {Ni(acac)_2_ + L^2^} was established. This indicates the unusual catalytic activity of the resulting triple {Ni(acac)_2_∙L^2^∙PhOH} complexes. Triple {Ni(acac)_2_∙L^2^∙PhOH} (L^2^ = NaSt) complexes have no analogs in the world in terms of activity and stability [23].

We hypothesized that the high efficiency and stability of heteroligand nickel complexes in the selective oxidation of ethylbenzene may be associated with the formation of supramolecular structures due to intermolecular hydrogen bonds and, possibly, other non-covalent interactions. We have proposed a new approach to studying the possibility of the formation of supramolecular structures in catalytic processes **–** using the method of atomic force microscopy (AFM). This article presents data from an AFM study of supramolecular structures based on Fe-acetylacetonate complexes with crown ethers and quaternary ammonium salts (oxidation catalysts, models Fe-ARD and Dke1) and model porphyrin complexes with PhOH and tyrosine (models of cytochrome P450-dependent monooxygenases and heme-dependent tyrosine hydroxylase).

## 2. Results

The enzymatic cleavage of C-C bonds in β-diketones is of interest in connection with various aspects of bioremediation, biocatalysis, and mammalian physiology. The mechanisms of C-C bond cleavage vary from hydrolytic processes involving metal compounds to processes catalyzed by dioxygenases [21,22,24,25,26].

It is known that β-diketo compounds that form chelates with metal ions are activated in relation to the electrophilic attack of various reagents. For example, these are reactions of addition of unsaturated electrophiles E to the γ-atom C of metal β-diketones. The limiting stage of such reactions is the formation of an intermediate resonance-stable zwitterion [M(L^1^)_n_^+^•E^−^], in which proton transfer occurs with the subsequent formation of reaction products [27]. Nickel complexes turned out to be the most effective in such reactions.

Previously, we assumed that the selective catalyst, Ni(acac)(AcO)·L^2^ may be formed as a result of the ligand L^2^-controlled outer sphere regioselective addition of O_2_ to a nucleophilic carbon atom (γ-C) of one of the acac^−^ ligands. The coordination of the electron-donating axial ligand L^2^ with the Ni(acac)_2_ complex, which promotes stabilization of the intermediate zwitterion [L^2^(acac)Ni(acac)+ ··· O_2_^−^], may lead to an increase in the probability of regioselective addition of O_2_ at the γ-carbon atom of the acetylacetonate ligand activated by coordination with the nickel(II) ion. Further inclusion of oxygen into the chelate acac ring accompanied by a proton transfer and the redistribution of bonds in the transition complex (the Criegee rearrangement) can lead to the rupture of the cyclic configuration and can end with the formation of the chelate ligand OAc−, acetaldehyde, and CO (Figure 1a,b):Ni(acac)_2_ L^2^ + O_2_ → [L^2^(acac)Ni(acac)^+^ ··· O_2_^−^] → L^2^(acac)Ni(AcO) + MeCHO + CO,2Ni(acac)_2_ L^2^ + O_2_ → Ni_2_(AcO)_3_(acac)L^2^ + 3MeCHO + 3CO + L^2^ (#)

The structure of the synthesized active supramolecular Ni_2_(AcO)_3_(acac)L^2^ (L^2^ = NMP) (“A”) complex was confirmed kinetically and using different physicochemical methods (mass spectrometry, electron, and IR spectroscopy) and elemental analysis. The composition of the Ni_2_(AcO)_3_(acac)L^2^ complex is supported by data on the analysis of acetaldehyde in the oxidation products of the complex (according to the equation marked with the sign #). It was established that complex “A” is a catalyst of the ethylbenzene oxidation into PEH [1]. The putative structure of complex “A” is presented in Figure 1b. The only known Ni-containing dioxygenase NiARD, which catalyzes the oxidative decomposition of *β*-diketones, operates in an analogous way [21].

The overall kinetics of ethylbenzene oxidation in the presence of catalytic Ni(or Fe)(II)L^1^_2_)*_x_*(L^2^)*_y_*complexes suggests that the formation of Fe-active complexes can also be the result of regioselective addition of ligand L^2^ to the γ-C-atom of the acetylacetonate ligand (controlled by ligand L^1^). We hypothesize that in this case, the oxidative transformation of Fe-heteroligand complexes may occur by a mechanism similar to the action of Fe-acetylacetone dioxygenase Dke1 [22] and FeARD [21]. This mechanism comprises an oxygen activation (Fe^II^ + O_2_ → Fe^III^ O_2_^•−^) step. Analogous with action of Dke1 and FeARD, the transformation of the Fe-complex may occur through the formation of an intermediate complex with a chelate ligand containing a 1,2-dioxetane moiety. The reaction ends with the formation of the chelate ligand OAc^−^ and methylglyoxal as the second decomposition product (Figure 2a), which leads to the formation of a complex of the proposed structure Fe*_x_*L^1^*_y_*(L^1^_ox_)*_z_*(L^2^)*_n_*•H_2_O_m_ (L^1^_ox_ = OAc^^−^^).

### 2.1. AFM Study of Supramolecular Structures Based on Ni-Heteroligand Complexes

Atomic force microscopy (AFM) is a widely used method for studying and characterizing various properties of biological, organic, and inorganic materials [28]. AFM contributed to the development of nanotechnology and is still used to study various nanomaterials such as carbon nanotubes and graphene.

In recent years, a research direction that has received significant development is “supramolecular chemistry on surface”. The concept of using supramolecular interactions, such as coordination bonds, halogen bonds, and hydrogen bonds, to control molecular organization on surfaces has been developed [29]. Supramolecular structures open up a promising direction for studying non-covalent interactions [30].

We have proposed the AFM method as a new approach for studying the role of hydrogen bonds and other non-covalent interactions and supramolecular structures in the mechanisms of catalysis by nickel and iron complexes, which are catalysts for the oxidation of hydrocarbons and models of active centers of enzymes [31].

It is known that heteroligand complexes are more active than homoligand complexes in reactions with electrophiles [1]. Thus, we assumed that the increase in the stability of the active M(II)_x_L^1^_y_(L^1^_ox_)_z_(L^2^)_n_(H_2_O)_m_ complexes, which are intermediate products of the transformation of the primary M(II)L^1^_2_L^2^ complexes, is apparently associated with the formation of stable supramolecular structures due to intermolecular hydrogen bonds and other non-covalent bonds. Based on the literature and our different physico-chemical data (see above), it could be assumed that the Ni_2_(OAc)_3_(acac)•NMP•2H_2_O complexes (“A”) are capable of forming macrostructures due to intermolecular hydrogen bonds (H_2_O – NMP, H_2_O − acetate (or acac–) group) and possibly other non-covalent interactions.

Indeed, using the AFM method, we recorded the self-organization of supramolecular complex “A” into supramolecular structures due, possibly, to hydrogen bonds and other non-covalent interactions (Figure 1b).

Three-component systems {{Ni(acac)_2_ + L^2^ + PhOH} (L^2^ = N-methyl-2-pyrrolidone (NMP), HMPA, MSt, M = Na, Li) are the most effective catalytic systems for the selective oxidation of ethylbenzene to PEH under the influence of O_2_.

Kinetic and spectral methods have proven the formation of triple Ni(acac)_2_∙L^2^∙PhOH complexes. Intra- and outer-sphere coordination of the activating ligands L^2^ = NMP and L^3^ (in the case of L^3^ = L-tyrosine) with Ni(acac)_2_ was recorded using UV spectroscopy [32]. Triple complexes are very stable and are not transformed by molecular O_2_ during the oxidation process for a long time. The AFM method revealed the self-organization of supramolecular structures based on Ni(acac)_2_∙L^2^∙PhOH (L^2^ = NaSt) with a maximum particle height of up to 300 nm [23]. These data indicate a high probability that similar supramolecular structures are formed under real conditions of the catalytic process. In this case, it seems that when discussing the high stability of the resulting structures, it is necessary to take into account the very probable cationic π-interactions [33,34].

The balance between the interactions of molecules with the surface and intermolecular interactions determines the formation of supramolecular structures on the surface. The AFM data we obtained do not mean that exactly such structures are formed in the real process. However, the existence of self-organization of supramolecular structures on the surface apparently indicates a high probability of the formation of analogous structures under real oxidation conditions.

Iron- and nickel-containing acireductone dioxygenases (ARDs) are enzymes involved in the methionine salvage pathway (MSP), a universal pathway for the conversion of sulfur-containing metabolites to methionine. The mechanism of action of ARD in relation to common substrates (1,2-dihydroxy-3-oxo-5(methylthio)pent-1-ene (β-diketone-acireductone) and molecular oxygen) is determined by the nature of the metal. Fe-ARD, similar to the action of Dke1, catalyzes the addition of O_2_ to the acireductone to form formate and 2-keto-4-(thiomethyl)butyrate, the precursor methionine. Ni-ARD catalyzes a pathway that does not lead to methionine, but Ni-ARD produces formate, carbon monoxide, and 3-methylthiopropionate (off pathway) [21,35] (Figure 2).

The (M(II,III)L^1^_n_)*_x_*∙(L^2^)*_y_*, M(II)*_x_*L^1^*_y_*(L^1^_o**x**_)**_z_**(L^2^)*_n_*(H_2_O)*_m_* (M = Ni, Fe) complexes are structural and functional models of the active sites of acireductone dioxygenases. Therefore, the results of AFM studies based on model complexes can be useful for explaining the mechanism of action of ARD enzymes. Thus, the decrease in Ni-ARD activity in the MSP cycle may be associated with the formation of multidimensional stable structures (analogous with structures based on heteroligand nickel complexes, Figure 1b).

### 2.2. AFM Study of Supramolecular Structures Based on Fe-Heteroligand Complexes

One of the important properties of crown ethers is selective complexation. Crown ethers are of interest to researchers as models of active sites of enzymes. Intermolecular and intramolecular hydrogen bonds play an important role in processes involving crown ethers.

It is known that crown ethers and R_4_NX are capable of catalyzing the electrophilic addition to the γ-C atom of an acetylacetonate ligand. The quaternary ammonium salts form complexes with acetylacetone in hydrocarbon media with strong hydrogen bonds R_4_N^+^(X∙∙∙HOCMeCHCOMe)^−^.

We used the ability of quaternary ammonium salts and crown ethers to form complexes with transition metal compounds to create effective catalysts for the selective oxidation of ethylbenzene to PEH. The role of H-bonds in the catalysis mechanism has been established when small amounts of water are added to the reaction catalyzed by {ML^1^_n_ + L_2_} systems (M = Fe, L^1^ = acac^−^, L^2^ = 18-crown-6 or R_4_NBr) [1].

By the AFM method, we studied the self-organization of iron complexes Fe(III)_x_(acac)_y_18C6_m_(H_2_O)_n_ into supramolecular structures due to hydrogen bonds and, possibly, other non-covalent interactions. Figure 2b shows a 3D AFM image of supramolecular structures, formed by applying the olution Fe(III)*_x_*(acac)*_y_*18C6_m_(H_2_O)_n_ complex to a hydrophobic modified silicone surface. As can be seen, the resulting structures, about 4 nm high (Figure 2b), resemble the cavities of cellular microtubules in shape (Figure 2c) [31].

The possibility of forming supramolecular nanostructures based on Fe(III)*_x_*(acac)*_y_*CTAB_m_, which is an active catalyst for the oxidation of ethylbenzene in PEH, was investigated using the AFM method. We applied solutions of Fe(III)*_x_*(acac)*_y_*CTAB_m_ in CHCl_3_ or H_2_O on the hydrophobic modified silicon surface. To reduce the probability of micellization, CTAB concentrations 5–10 times lower than the Fe(acac)_3_ concentrations were used.

Figure 3 shows 3D and 2D AFM images of structures based on the Fe(III)*_x_*(acac)*_y_*CTAB_m_(CHCl_3_)_p_ complex. As can be seen, these nanostructures are similar to structures based on Fe(III)*_x_*(acac)*_y_*18C6*_m_*(H_2_O)*_n_* and resemble the shape of the cavity of cellular microtubules (Figure 2b), but with a less pronounced structure. The height of the particles is about 12–13 nm. UV spectroscopy data indicated outer-sphere coordination of various R_4_NX (including CTAB) with Fe(acac)_3_ (hydrogen bonds) [1,31]. In this case, in addition to hydrogen bonds, as well as other non-covalent interactions, cation-π interactions should probably be taken into account [34].

When applying aqueous solutions of Fe(III)_x_(acac)_y_CTAB_m_ to the hydrophobic surface of the modified silicon, we observed particles that may be remnants of micelles. Figure 4a,b shows 2D AFM images of structures based on iron Fe(III)*_x_*(acac)*_y_*CTAB_p_(H_2_O)_q_ complexes, obtained by applying an aqueous solution of the complex to a hydrophobic modified silicon surface. The particles in Figure 4, similar to micelles, are probably due to the rapid evaporation of the water necessary for their existence, very unstable, and quickly destroyed (even during measurements).

### 2.3. Putative Role of the Tyr Fragment in the Enzyme’s Action

Earlier, we established the phenomenal effect of phenol in the oxidation of ethylbenzene by oxygen, catalyzed by nickel complexes [23]. We proposed a regulatory role for the Tyr fragment located in the second coordination sphere of NiARD in the mechanism of enzyme action [36]. The Tyr fragment can inhibit the action of NiARD, similar to the proposed involvement of the Tyr fragment in reducing the activity of Homoprotocatechuate 2,3-dioxygenase [37]. Using the AFM method, we observed the formation of stable supramolecular structures based on triple Ni(acac)_2_∙L^2^∙L^3^ (L^2^ = NMP, HMPA, His; L^3^ = PhOH, Tyr) complexes, also including the amino acids L-histidine (L^2^ = His) and L-tyrosine, which can be considered as the primary model of NiARD [31,32]. These facts support the putative role of the Tyr fragment as a regulatory factor that reduces NiARD activity.

The AFM method was used for the first time to demonstrate the possibility of self-organization into supramolecular structures due to hydrogen bonds of Fe(III)*_x_*(acac)_y_(His)*_m_*(Tyr)*_n_*(H_2_O)*_p_* complexes, which model the active center of FeARD. We observed self-organization into tubulin-like structures, as in the case of the model complexes Fe(III)*_x_*(acac)*_y_*18C6_m_(H_2_O)_n_ and Fe(III)*_x_*(acac)*_y_*CTAB_m_(CHCl_3_)_p_. The formation of tubulin-like structures may facilitate the regioselective addition of activated oxygen to the acereductone ligand and subsequent reactions leading to methionine formation.

Tyrosine residues are capable of participating in hydrogen bonding, conformation, and molecular recognition. Tyrosine is important in enzymatic reactions.

For example, it is assumed in the model of Methyltransferases and structural surveys that methyl CH•••O hydrogen bonding (with participation of Tyr fragments) represents a feature of AdoMet-dependent Methyltransferases as the universal mechanism for methyl transfer [36].

Non-heme tyrosine hydroxylase is found in the central nervous system. This iron monooxygenase catalyzes the hydroxylation of tyrosine to L-3,4-dihydroxyphenylalanine. Tyrosine, tetrahydrobiopterin, and O_2_ take part in the catalysis [38].

Heme-dependent l-tyrosine hydroxylases (TyrHs) constitute a new enzyme family in contrast to the non-heme iron enzymes [39,40]. Typical TyrHs exhibit dual reactivity, cleaving CH and CF bonds upon exposure to 3-fluoro-1-tyrosine (3-F-Tyr) as a substrate. A mechanistic study shows that nature uses Cpd I as an oxidizing agent [40]. The TyrH enzyme has great potential in biocatalysis as it is able to activate both C-H and C-F bonds in a regioselective manner in aromatic substrates Figure 5 [39]:

Our AFM studies of model systems indicate the participation of tyrosine and histidine as one of the possible regulatory factors in the functioning of enzymes of the P450 family and the heme-dependent tyrosine hydroxylase TyrH. The common intermolecular interaction in nature is the binding of Porphyrins due to H-bonds. An example of the simplest self-assembling supramolecular porphyrin systems is the formation of a dimer based on a carboxylic acid group [12,13].

Figure 6 shows AFM images of nanostructures based on natural Hemin and systems including Hemin: {Hem + PhOH + His} and {Hem + Tyr + His} (Hem = Hemin) (after applying solutions to the modified silicon surface). As can be seen from Figure 6, the supramolecular structures based on only Hem in the absence of ligand additives differs in height and shape () (Figure 6a,b) from the structures based on the {Hem•L^3^•His} (L^3^ = PhOH or Tyr) complexes (Figure 6c,e, **h = 20 nm**). The observed structure is due to the intermolecular interactions of Hem and of {Hem•L^3^•His} complexes due to hydrogen bonds and, possibly, other non-covalent bonds in the process of spontaneous self-organization on the surface. H-bonds NH•••O or N•••HO and π–π- intermolecular interplanar interactions possibly participate in the formation of supramolecular structure based on {Hem•L^3^•His} complexes.

## 3. Discussion

The organization of several catalytic centers through noncovalent interactions into a specific, cooperative framework was supposed to increase reactivity, selectivity, and efficiency, similar to the action of real enzymes.

In catalytic processes, typically the primary catalytic complexes are less active than the actual catalysts formed during the process. It has been established that the true catalysts are heteroligand Ni(Fe)*_x_*L^1^*_y_*(L^1^_ox_)*_z_*(L^2^)*_n_*•H_2_O_m_ complexes, which are intermediates of the outer-sphere oxidative transformation of primary M(II)(L^1^_2_)*_x_*(L^2^)_y_ complexes and are more effective in the oxidation of ethylbenzene to PEH compared to primary complexes. It turned out that the mechanism we established for the formation of Ni(Fe)*_x_*L^1^*_y_*(L^1^_ox_)*_z_*(L^2^)*_n_*•H_2_O_m_ catalysts is similar to the action of Ni,Fe-acireductone dioxygenase Ni(Fe)ARD and Fe-acetylacetone dioxygenase Dke 1. We hypothesized that the high stability of the active forms of ethylbenzene oxidation catalysts may be associated with the formation of stable supramolecular structures due to intermolecular hydrogen bonds and, possibly, other non-covalent interactions. This hypothesis was confirmed using the AFM method [31].

The results of AFM study model complexes may be useful to explain the mechanism of action of these enzymes. The formation of outer-sphere H-bonds and multimeric forms due to H-bonds may be one possible way to regulate the activity of the two Ni(Fe)ARD enzymes in the MSP cycle. In the case of FeARD, the self-organized supramolecular structures resembling the shape of the tubulin microfiber cavity (we observed in model Fe-acetylacetonate complexes with 18C6 and CTAB) may promote O_2_ activation and reactions leading to methionine formation.

Using the AFM method, convincing evidence was obtained in favor of the participation of the tyrosine fragment in the mechanism of action of Ni(Fe)ARD. The formation of stable supramolecular structures based on ternary Ni(acac)_2_∙L^2^∙L^3^ (L^2^ = NMP, HMPA, His; L^3^ = PhOH, Tyr) complexes—the primary model of NiARD—confirms the putative role of the Tyr fragment as a regulatory factor of NiARD activity [23]. In the case of FeARD catalysis, the formation of methionine can be facilitated by the self-organization of iron complexes with the participation of the Tyr fragment into structures resembling tubulin microtubules in shape [31].

We have observed the self-organization of model systems {Hem + Tyr(PhOH) + His} (Hem = Hemin) into structures that are different in shape and size from Hem-based supramolecular structures. The obtained results indicate the participation of tyrosine and histidine fragments as one of the possible regulatory factors in the functioning of heme-dependent TyrHs [39,40] and enzymes (heme proteins) of the P450 family.

In addition to the enzymatic function of the ARD enzyme, some studies suggest other functions of ARD [35]. For example, carbon monoxide CO, which is one of the products of NiARD function, is an antiapoptotic molecule in mammals. The human enzyme regulates the activity of matrix metalloproteinase I (MMP-I). Some studies have shown that ARD in humans suppresses cancer development [41]. It was found that the mixed-ligand [Ni(L^1^)_2_L^2^]H_2_O (L^1^ = acac^−^, L^2^—2-aminopyridine) complex has high antitumor activity in vitro against the gastric cancer cell line MKN45 [42].

## 4. Materials and Methods

In the AFM study, we used the scanning probe microscope SOLVER P47 SMENA10 (Adm. Distr. Zelenograd of Moscow, 124490, Russia), using an NSG30 cantilever with a radius of curvature of 10 nm, tip height of 10–15 µm, and cone angle of ≤ 22° in tapping mode on a resonant frequency 150 KHz.

We used an NSG30_SS cantilever (Nanosensors^TM^ Advanced Tec^TM^ AFM probes, CH-2000 Neuchatel, Switzerland) with a radius of curvature of 2 nm, a resonance frequency of 300 kHz, and a force constant of 22–100 N/m. TipsNano in tapping mode used for the AFM research of supramolecular structures.

As the substrate, the polished Silicone surface that was specially chemically modified was used.

Waterproof modified Silicone surface was employed for the self-assembly-driven growth due to H-bonding of complexes with the Silicone surface. Saturated solution complexes in chloroform (CHCl_3_) or in H_2_O were used. The solutions were as follows: solutions of complexes of Fe^III^*_x_*(acac)*_y_*CTAB*_m_* (CHCl_3_ or H_2_O) and Fe^III^*_x_*(acac)*_y_*18C6*_m_* (H_2_O); solutions of {Hem•L^3^•His} (L^3^ = PhOH) (in CHCl_3_); and solutions of the complex {Hem•L^3^•His} (L^3^ = Tyr) in mixture {C_2_H_5_OH+H_2_O (1:1)}. Solutions were placed on a surface, maintained for some time, and then solvent was deleted from the surface by means of a special method—the spin-coating process.

UV spectroscopy was used to prove the outer-sphere coordination QX with Fe(acac)_3_ (CHCl_3_ solution). Quartz cuvettes, which were 1 mm thick, were used to record the spectra in the UV regions. The spectra were recorded on a high-sensitivity spectrophotometer, “UV-VIS-SPECS”.

## 5. Conclusions

Supramolecular chemistry is an area of research that has developed significantly in recent years. Studies of enzymes have shown that the coordination spheres of the active centers of the enzyme play a decisive role in determining the properties of the metal factor. We have proposed a new approach to assessing the role of hydrogen bonds and supramolecular structures in the mechanisms of homogeneous and enzymatic catalysis—the AFM method.

Using AFM, we observed self-organization due to intermolecular H-bonds and possibly other non-covalent interactions of supramolecular structures based on metal complexes, which are structural and functional models of the active sites of enzymes of the P450 family, heme-dependent l-tyrosine hydroxylases (TyrHs), Ni(Fe)ARD. The obtained AFM data are of interest from the point of view of assessing the possible role of outer-sphere regulatory interactions in the mechanisms of homogeneous and enzymatic catalysis. The fact that such assembly takes place, and the fact that this type of interaction in the active center of the enzyme cannot be ignored, is evidenced by the images that we see in examples of complexes that are models of the active centers of enzymes.

## Data Availability

All of the experimental data presented belong to the authors of this manuscript and are available.

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
