# Peer review of "Formation of Supramolecular Structures in Oxidation Processes Catalyzed by Heteroligand Complexes of Iron and Nickel: Models of Enzymes"

_ijms, 2025, doi:10.3390/ijms26168024_

Round 1
Reviewer 1 Report
Comments and Suggestions for Authors
The authors present a series of work that probes the supramolecular structure of complexes by AFM. As the authors stated this is a novel perspective to investigate the interactions between the complexes. This manuscript should be almost ready for publication except for a few minor points for the authors to consider.
In the mechanism part in Results before section 2.1, in Scheme 1a, do the authors have evidence to eliminate the possibility of alternate pathway? For example, some porphyrin complexes were shown to catalyze oxidation at the meso position of the ligands. Some complexes also catalyze oxidation at the 3 position of acetylacetone. So could the first step of the reactions be formation of a ketone group or bis-hydroxy? Also, should not the acac ligand be symmetric, i.e., the 2,3,4 carbons would form conjugated bonds and there would not be a difference like "C-C" and "C=C" bonds? Could different pathways be favoured in the cases of Schemes 1 and 2?
I also recommend increasing the font size in Scheme 1a and 2a.
Have the authors considered checking any order on the magnitude of a few nanometers by SEM or SAXS, for example?
Author Response
Responses to reviewer 1.
The article ijms-3788247:
Formation of Supramolecular Structures in Oxidation
Processes Catalyzed by Heteroligand Complexes of Iron and Nickel: Models of Enzymes
Ludmila I. Matienko, Elena M. Mil, Anastasia A. Albantova and Alexander N. Goloshchapov
The authors would like to thank Reviewer 1 for reading our article in detail and making valuable comments, which certainly contribute to the quality of our article and will be useful in the future.
Please note that the structure of the article has been changed in accordance with the editors’ comments and the comments of reviewer 2. Figures 2 and 3 have been removed from the Introduction. Part of the text from the Results has been moved to the Introduction. Discussion and Conclusion have been shortened. All changes in the text of the article are marked in yellow.
Comments:
The authors present a series of work that probes the supramolecular structure of complexes by AFM. As the authors stated this is a novel perspective to investigate the interactions between the complexes. This manuscript should be almost ready for publication except for a few minor points for the authors to consider.
Comment 1:
In the mechanism part in Results before section 2.1, in Scheme 1a, do the authors have evidence to eliminate the possibility of alternate pathway? For example, some porphyrin complexes were shown to catalyze oxidation at the meso position of the ligands. Some complexes also catalyze oxidation at the 3 position of acetylacetone. So could the first step of the reactions be formation of a ketone group or bis-hydroxy? Also, should not the acac ligand be symmetric, i.e., the 2,3,4 carbons would form conjugated bonds and there would not be a difference like "C-C" and "C=C" bonds? Could different pathways be favoured in the cases of Schemes 1 and 2?
Response to comment 1:
In Schemes 1 and 2 we propose mechanisms as the most likely ones to explain our experimental data (for nickel complexes the data are given in the Results after the description of the mechanism).
Taking into account your comments, we have added to the description of the mechanisms (see the beginning of the Results).
In this work, we discuss the reaction of oxygen only with the acac‾ ligand activated as a result of coordination with a metal (nickel, iron). Similar changes in b-diketonate ligands in mixed-ligand metal complexes under the influence of O2 are described in the mechanism of action of NiARD (acireducton dioxygenase), in oxygenation reactions that imitate the action of dioxygenases (quercetinase (reference [26])) (Scheme 1a).
As for Fe-heteroligand complexes, in this case oxidative transformation can occur by a mechanism similar to the action of Fe-acetylacetone dioxygenase Dke1 and FeARD (Fe-acireductone dioxygenase), which is consistent with our experimental data (Scheme 2a). .
In the future, we intend to consider outer-sphere reactions of porphyrin complexes.
Comment 2:
I also recommend increasing the font size in Scheme 1a and 2a.
Response to the comment 2:
We took your comment into account and increased the sizes of the Schemes and font (Schemes 1a and 2a)
Comment 3:
Have the authors considered checking any order on the magnitude of a few nanometers by SEM or SAXS, for example?
Response to the comment 3:
In our work, we used only the possibilities provided by the AFM method. We were interested in the possibility of the formation of nanostructures based on catalytic active heteroligand complexes of nickel and iron on the surface due to intermolecular non-covalent bonds.
We will follow your recommendation to use methods (SEM or SAXS) to evaluate nanoscale structures.
Reviewer 2 Report
Comments and Suggestions for Authors
The manuscript entitled “Formation of Supramolecular Structures in Oxidation Processes Catalyzed by Heteroligand Complexes of Iron and Nickel. Models of Enzymes” deals with self-organized supramolecular structures and their properties using AFM microscopy and UV spectroscopy. The authors for the first time proposed a new approach to study the role of H-bonds and supramolecular structures in the mechanisms of homogeneous and enzymatic catalysis - the AFM method. The results are quite useful, but the presentation of the material should be corrected. Here are my comments on the manuscript:
- It is recommended to avoid citing literature in the “Abstract” section.
- Lines 107 and 111. There is no need to provide the same citation [1] in one paragraph. Besides, the names of the authors should not be provided – they can be found in the “References” section. Furthernore, the ref [1] is cited many times - it should be described whether it is a previous research made by the authors or the literature review.
- Lines 141-187 appear to be as a part of “Introduction” section rather “Results”. This should be rewritten.
- Scheme 1c quality should be improved to enhance readability. The same for Figure 8b.
- The size of molecular structures in Scheme 1a and 2a should be increased to enhance readability.
- In general, the Results and Discussion sections should be revised to be presented more concise.
Author Response
Responses to reviewer 2.
The article ijms-3788247:
Formation of Supramolecular Structures in Oxidation
Processes Catalyzed by Heteroligand Complexes of Iron and Nickel: Models of Enzymes
Ludmila I. Matienko, Elena M. Mil, Anastasia A. Albantova and Alexander N. Goloshchapov
The authors would like to thank Reviewer 2 for reading our article in detail and making valuable comments, which certainly contribute to the quality of our article and will be useful in the future.
Please note that the structure of the article has been changed in accordance with the editors’ comments and the comments of reviewer 2. Figures 2 and 3 have been removed from the Introduction. Part of the text from the Results has been moved to the Introduction. Discussion and Conclusion have been shortened. All changes in the text of the article are marked in yellow.
Comments:
The manuscript entitled “Formation of Supramolecular Structures in Oxidation Processes Catalyzed by Heteroligand Complexes of Iron and Nickel. Models of Enzymes” deals with self-organized supramolecular structures and their properties using AFM microscopy and UV spectroscopy. The authors for the first time proposed a new approach to study the role of H-bonds and supramolecular structures in the mechanisms of homogeneous and enzymatic catalysis - the AFM method. The results are quite useful, but the presentation of the material should be corrected. Here are my comments on the manuscript:
Comment 1:
It is recommended to avoid citing literature in the “Abstract” section.
Response to comment 1:
We considered your comment and removed citing literature from “Abstract”.
Comment 2:
Lines 107 and 111. There is no need to provide the same citation [1] in one paragraph. Besides, the names of the authors should not be provided – they can be found in the “References” section. Furthernore, the ref [1] is cited many times - it should be described whether it is a previous research made by the authors or the literature review.
Response to comment 2:
We took your comment into account and made appropriate changes to the text.
Comment 3:
Lines 141-187 appear to be as a part of “Introduction” section rather “Results”. This should be rewritten.
Response to comment 3:
We already wrote at the beginning of the responses to comments that part of the Results was moved to the Introduction. However, we left the description of Schemes 1a and 2a in the Results. In connection with the comments of reviewer 2, it became necessary to consider them in more detail.
Comment 4:
Scheme 1c quality should be improved to enhance readability. The same for Figure 8b (now Fig. 6b).
Response to comment 4:
We took your comment into account and made appropriate changes to the Scheme 1c and Figure 6b.
Comment 5:
The size of molecular structures in Scheme 1a and 2a should be increased to enhance readability.
Response to comment 5:
To enhance readability we made appropriate changes to the Scheme 1a and 2a.
Comment 6:
In general, the Results and Discussion sections should be revised to be presented more concise.
Response to comment 6:
We took your comment into account. Results, Discussion and Conclusion have been shortened
Round 2
Reviewer 2 Report
Comments and Suggestions for Authors
The authors have addressed all the comments. I have no additional questions.